# Adoption of Sustainable Agriculture Practices through Participatory Research: A Case Study on Galapagos Islands Farmers Using Water-Saving Technologies

**DOI:** 10.3390/plants11212848

**Published:** 2022-10-26

**Authors:** Patricia Jaramillo Díaz, Anna Calle-Loor, Ekaterina Gualoto, Carlos Bolaños, David Cevallos

**Affiliations:** 1Charles Darwin Research Station, Charles Darwin Foundation, Puerto Ayora 200105, Ecuador; 2Facultad de Ciencias, Universidad de Málaga, 29010 Málaga, Spain; 3Institute of Botany and Landscape Ecology, University of Greifswald, 17489 Greifswald, Germany

**Keywords:** productivity traits, Groasis Waterboxx^®^, individual fruit weight, wet-warm, dry-cold season, Santa Cruz, Floreana, maximum plant height

## Abstract

Agriculture in the populated islands of the Galapagos Archipelago, a protected area due to its unique biodiversity, has been detrimental to its conservation but highly required to meet food necessities. A potential solution to make agricultural farming more sustainable is adopting water-saving technologies (WSTs). Therefore, this study aimed to test the effectiveness of using WSTs such as Groasis Waterboxx^®^ in three of the most valuable crops in the islands through participatory research with the involvement of a group of farmers from the Floreana and Santa Cruz islands and explore a possible transition to more sustainable agricultural practices. *Capsicum annuum*, *Cucumis sativus* and *Solanum lycopersicum* were cultivated using Groasis Waterboxx^®^ and compared to conventional irrigation practices (drip-irrigated controls) to assess the variability of productivity, the number of fruits and individual fruit weight (IFW). In addition, differences in plant traits were analyzed by crop, and island. Results suggested that WSTs such as Groasis Waterboxx^®^ may provide on-farm benefits regarding the yields of the studied traits. From this study, it is difficult to determine whether participation in such a research study will permanently change irrigation practices. However, the participant’s responses to the study suggest an increase in their understanding of the use and benefits of WST.

## 1. Introduction

In recent decades, several researchers have highlighted the urgency of knowledge, innovations and actions to promote sustainable transitions [1,2,3]. For example, water-saving technologies (WST) in agricultural systems are a sustainable farming strategy [4,5]. WSTs are essential sustainability tools that supply water directly to the plant roots, reducing water runoff while maintaining or even increasing crop productivity [5]. These technologies might be valuable in the Galapagos Islands, where water for irrigation is scarce, and there is an urgent need to satisfy local food production for the population.

The Galapagos Archipelago includes 123 islands, some with high levels of endemism and most with unique biodiversity [6]. The Galapagos Islands have been considered a UNESCO natural world heritage site since 1978 (International Union for Conservation of Nature and Natural Resources). There are four populated islands: Santa Cruz, Floreana, Isabela and San Cristobal. The rapid population growth of these islands has resulted in issues with freshwater access and the use of brackish water for irrigation [7,8,9].

Water scarcity has been a constraint for the socio-economic development of Galapagos inhabitants. In recent decades, local growth and the tourism boom have severely threatened this resource [10]. In addition, climatic conditions produce high evapotranspiration and unequal rainfall in the agricultural zone of the populated islands.

Water scarcity occurs in yearly cycles. During the warm-wet season (December–June), characterized by heavy rainfall and evapotranspiration [11], the migration of the intertropical convergence zone controls rainfall [12]. During the dry-cold season (July–November), the draw on the water supplies grows enormously, and some farmers rely on external inputs of poor-quality water for irrigation. Moreover, soils are poorly developed and have low water-holding capacity [13].

Additionally, pepper (*Capsicum annuum* L.), cucumber (*Cucumis sativus* L.) and tomato (*Solanum lycopersicum* L.); called hereafter “*Capsicum*”, “*Cucumis*” and “*Solanum*”, respectively, are essential vegetable crops grown extensively in the Galapagos Archipelago, based on conversations with the local farmers. The performance of these crops is affected by many factors, with drought being the most critical, directly increasing food insecurity. Maintaining an adequate hydric status during the growth cycle is especially important in sensitive crops such as *Solanum* and *Capsicum*, which could lose turgor and form cracks at the surface, ending in rejections and non-marketable fruits.

Under these conditions, water scarcity is prevalent, leading to a dependency on brackish water to irrigate crops and inefficient and costly practices. Farmers are forced to buy brackish ground/crevice water from privately owned tanker trucks to irrigate crops, and small-scale subsistence farmers must transport water to planting sites by carrying it themselves.

Given that agriculture is an important sector for the economic development of Galapagos and given the current challenges facing Galapagos farmers, adopting sustainable practices is urgent to meet local food production for the population while reducing the impact on the unique ecosystems of the islands. The question arises as to how to engage farmers in a sustainable transition.

Advancing sustainable agriculture and developing innovations to achieve this end requires more inclusive ways of knowing and doing that bring together the concerns and perceptions of the various actors [14]. Therefore, the role of farmer participation in raising awareness, acquiring knowledge, implementing and adopting innovations for sustainable agriculture in rural areas cannot be underestimated [15]. 

In this regard, Aare and colleagues emphasize the potential of participatory research as an extension to mobilize farmers into action for sustainability [1]. This method aims to bridge the gap between research and practice through farmers’ active and voluntary participation as partners in the research process, who contribute their time, effort and skills to generate data for a scientific project [16,17]. Participatory research can achieve two primary objectives: first, expanding the potential for collecting, analyzing and preserving information; second, engaging and motivating the community to get involved [16]. In addition, technology transfer is considered more effective when farmers directly test a technology [18,19].

However, it is crucial to consider that the ability of participants to benefit from participatory research projects directly mediates their motivation to engage in the project. In that sense, when involving farmers in participatory research to adopt more sustainable practices, it is particularly relevant to benefit them directly. Addressing issues related to their needs, such as, irrigation and crop yield, and generating relevant solutions by testing alternatives on the farm, enables them to innovate and adapt [1,20,21]. 

This study aimed to test the effectiveness of the water-saving technology (WST), Groasis Waterboxx^®^, on three important crops in the Galapagos Islands through participatory research as a case study to explore a possible transition to sustainable agricultural practices.

## 2. Results

### 2.1. Participation of Farmers

We first approached potential participants, explaining the project’s scope and inviting farmers to participate. During this first outreach, the research team perceived an initial rejection response. Farmers were defensive about the possibility of participating in the experiment and expressed concern that they would not directly benefit from such collaboration. Given that the Galapagos Islands are a world-famous site for the study of evolution and conservation, local people are constantly exposed to the presence of researchers who, in many cases, require the participation of local people in their studies. Participants in this study claimed they were tired of being used as objects of study and that farmers want answers and solutions to urgent daily problems, such as the eradication of pests and invasive species, which significantly impact agricultural production. However, despite their initial rejection, some farmers agreed to receive the research team on their farms for an individual visit, which allowed a one-to-one conversation to provide farmers with more detailed information about the WSTs, and explain their participation in the project. After learning about the use of WST, farmers were curious about the benefits of adopting these new technologies. In particular, they were interested in the potential of these technologies to reduce the use of brackish water and improve the productivity of their crops.

The first activity in the experimental process was crop selection. The participant’s farming knowledge and experience informed their decision. For instance, farmers in Floreana, an island with a smaller population of ~150 inhabitants, have smaller and more limited croplands and resources, which makes them more hesitant to accept research interventions since these could risk their production. Consequently, to reduce the risk of crop failure, they decided to work with *Cucumis* and *Solanum* due to the outstanding productivity of these crops on the island. Participants in Santa Cruz selected three crops for the experiment: *Capsicum*, *Cucumis* and *Solanum* due to their significant commercial value and relevance to their livelihoods.

During the following stages of the experimental process, some challenges arose due to the farmer’s busy schedules since they had to respond to the diverse needs of their farms to generate an income. For many, it was not easy to perform regular monitoring. Moreover, as agricultural production is the source of their livelihood, farmers found it difficult not to intervene when the plants were ill or dying, which meant lower production. Despite these challenges, the farmers’ collaboration and commitment to the project helped to collect and register consistent data and to detect and correct specific problems, such as clogging of the wick in the Groasis Waterboxx^®^ by roots from different plants searching for water supply on the wick. At the end of the experiment, farmers expressed interest in further experimentation with WSTs in short-cycle crops. Nonetheless, since the agriculture sector in Galapagos currently does not offer adequate profitability [22] they perceived economic constraints as the main obstacle to expanding the use of WSTs due to the elevated cost of these technologies. Notably, during the regular conversations with the farmers, the allusions to a lack of support from the local government and the need for training were common.

### 2.2. Effect of Water-Saving Technologies on the Selected Crop Traits

We compared the effectiveness of the WST Groasis Waterboxx^®^ versus conventional irrigation (control) in three crucial crops for farmers Capsicum, Cucumis and Solanum in two inhabited islands (Santa Cruz and Floreana), in three traits: productivity, number of fruits and individual fruit weight (detailed information is explained further). Additionally, we added the differentiated productivity during the two seasons of the year: wet-warm season (January to June) represented as “Wet” and dry-cold season (July to December) represented as “Dry”. Table 1 displays statistically significant values in bold.

#### 2.2.1. Plant Productivity

We considered plant productivity as the weight of the total number of fruits a plant produces during its life. The water-saving technologies have different effects based on the crop, island and season (dry-cold or wet-warm) (Table 1) due to the difference in precipitation and temperature in both seasons from 2016 to 2018 (Appendix A). 

We found significant differences between treatments on plant productivity in *Capsicum*, *Cucumis and*
*Solanum* (χ^2^ = 42.460, *p* < 0.001, χ^2^ = 4.006, *p* = 0.045 and χ^2^ = 6.336, *p* = 0.012, respectively) in Santa Cruz during the dry season (Table 1, Figure 1).

The crop productivity in the dry-cold season was 33% higher than in the wet-warm season (Appendix A). 

The productivity with Groasis Waterboxx ^®^ was 5% higher than control in the wet season, whereas control was 10% higher than Groasis Waterboxx^®^ in the dry season (Appendix A).

On Santa Cruz Island, in the wet season, *Capsicum* and *Solanum* had 5 and 6% higher productivity with Groasis Waterboxx^®^ than control, whereas in *Cucumis*, control was 7% higher than Groasis Waterboxx^®^. However, in the dry season, control had better productivity in *Capsicum* and *Cucumis* (32 and 18%), higher than Groasis Waterboxx^®^. In contrast, Groasis Waterboxx had better performance in *Solanum*, 10% higher than control (Appendix A). 

On Floreana Island, in the wet season, there was no productivity for *Cucumis* in none of the treatments. In contrast, there was only productivity in *Solanum* for Groasis Waterboxx^®^ but not for control. However, in the dry season, the productivity was higher in *Cucumis* and *Solanum* (20 and 17%) in Groasis Waterboxx^®^ than in control (Appendix A).

#### 2.2.2. Number of Fruits

We consider this parameter the total number of fruits that the plants produce in their productive time.

We found significant differences between treatments on the number of fruits in *Capsicum* in the dry season and *Solanum* in the wet season (χ^2^ = 41.863, *p* < 0.001; χ^2^ = 19.183, *p* < 0.001) in Santa Cruz (Table 1, Figure 1), and *Cucumis* in the dry season (χ^2^ = 9.344, *p* < 0.022) in Floreana (Table 1, Figure 2). 

The number of fruits was 49% higher in the dry season than wet season, considering all crops (Appendix A).

The number of fruits with Groasis Waterboxx ^®^ was 13% higher than control in the wet season, whereas control was 17% higher than the WST in the dry season (Appendix A).

On Santa Cruz Island, in the wet season, *Capsicum* and *Cucumis* had a similar number of fruits in both treatments. In *Solanum*, Groasis Waterboxx^®^ was 50% higher than control. On the other hand, in the dry season, *Capsicum* and *Cucumis* had a higher number of fruits (29% and 9%) using Control than the WST.

On Floreana Island, the number of fruits under control treatment in the dry season was higher in *Cucumis* and *Solanum* (40% and 25%) than in Groasis Waterboxx^®^ (Appendix A).

#### 2.2.3. Mean Individual Fruit Weight (IFW) 

The mean of IFW was considered in crops that produce more than one fruit per harvest. We found significant differences between treatments on average IFW in *Cucumis* and *Solanum* on both islands, Santa Cruz (χ^2^ = 22.567, *p* < 0.001, and χ^2^ = 8.2463, *p* < 0.001; Table 1, Figure 1), and Floreana (χ^2^ = 41.143, *p* < 0.001, and χ^2^ = 9.8319, *p* = 0.001; Table 1, Figure 2).

The IFW in the wet season was 12% higher than in the dry season (Appendix A).

Control was 8% higher than Groasis Waterboxx^®^ in the wet season, whereas the WST was 10% higher than control in the dry season (Appendix A). 

On Santa Cruz Island, in the wet season, *Cucumis* and *Solanum* had higher IFW (7 and 10%) using the control treatment, whereas, in the dry season, IFW was 10% higher with control in *Cucumis*, and 7% higher with Groasis Waterboxx^®^ in *Solanum*.

On Floreana Island, in the wet season, there was no IFW productivity for *Cucumis* in none of the treatments. In contrast, there was only IFW productivity in *Solanum* for Groasis Waterboxx^®^ but not for Control (Appendix A). On *Cucumis*, Groasis Waterboxx^®^ had 55% higher IFW than Control in the dry season. On *Solanum*, Groasis Waterboxx^®^ had 38% higher IFW than Control in the dry season (Appendix A). 

#### 2.2.4. Effect of the Research Year on Treatments

When we analyzed the effect of each year in productivity by treatment, we obtained statistical differences between years in control (χ^2^ = 168.07, *p* < 0.001) and in Groasis Waterboxx^®^ (χ^2^ = 168.07, *p* < 0.001), as shown in Figure 3.

Between treatments, the productivity in 2016 was 57% higher in Groasis Waterboxx than in control. In 2017 Groasis Waterboxx^®^ was 2% higher than control; however, in 2018, Control productivity was 39% higher than Groasis Waterboxx^®^ (Figure 3).

Within treatments, in the Control, the highest productivity (2018) was 63% higher than the lowest (2016), whereas, in Groasis Waterboxx^®^, the highest productivity was 27% than the lowest (2017), as we observe in Figure 3.

#### 2.2.5. Effect of the Treatments on Maximum Plant Height

The effect of Treatments (Control and Groasis Waterboxx^®^) on the maximum plant height reached by the plant differed statistically in *Cucumis* in Santa Cruz (χ^2^ = 10.801, *p* = 0.001), in *Solanum* in Santa Cruz (χ^2^ = 13.655, *p* < 0.001) and *Solanum* in Floreana (χ^2^ = 5.297, *p* = 0.021; Figure 4) 

The maximum plant height with the Control treatment was 6% higher than Groasis Waterboxx^®^ in *Capsicum*, 35% higher in *Cucumis* in Santa Cruz, and 10% higher in *Solanum* in Santa Cruz. In contrast, the maximum plant height with Groasis Waterboxx^®^ was 10% than Control in *Cucumis* in Floreana and 19% higher in *Solanum* in Floreana (Appendix A).

#### 2.2.6. Correlation between Maximum Plant Height and Total Productivity

The Pearson’s correlation between the maximum plant height reached by the plant and the total productivity of the plant during its life time shows a high correlation in *Capsicum* in both treatments (Waterboxx: R = 0.58, *p* < 0.001; control: R = 0.71, *p* < 0.001), in *Cucumis* in control only (Waterboxx: R = 0.12, *p* = 0.54; control: R = 0.52, *p* = 0.007) and a high correlation in *Solanum* in both treatments (Waterboxx: R = 0.36, *p* < 0.001; control: R = 0.27, *p* < 0.001, Figure 5).

## 3. Discussion

This section analyzes the results of the participatory exploratory phase regarding the effectiveness of water-saving technologies, and reflects on whether this process can foster WST adoption and transition to new sustainable agricultural practices based on the results from the present study.

There is little information available on the impact of drought stress on crop yield under Archipelago conditions. However, data from this study indicated that *Solanum*, *Capsicum* and *Cucumis* benefit from water-saving technologies, helping farmers sustain these crops’ sustainable production. 

We observed that the performance of the water-saving technologies varies between crops, seasons and islands. In productivity, we found significant differences between both treatments (Groasis Waterboxx^®^ and control) but only in the dry season. The majority of the harvesting period happened in the dry season; thus, the most important period for productivity in the studied crops (*Capsicum*, Cucumis and Solanum) occurred in the dry-cold season (July to December). For these three crops, the harvest period occurs 9 to 15 weeks after plantation [23]. Without irrigation, local farmers in tropical areas tend to sow the plants in the wet-warm period (January–June) and harvest in the dry-cold season to take advantage of the higher precipitation [22]; however, if farmers count on irrigation, they can sow anytime in the year [24]. Therefore, we could observe that productivity was higher in the dry season.

The productivity with Groasis Waterboxx^®^ was higher in the wet season than in the control. This is because the WST not only holds water but also collects rainwater and provides a harbor to the plants protecting them from extreme temperature conditions, making water use more efficient, reflected in the final productivity. On the other hand, in the dry season, control treatment based on drip irrigation was higher than the WST. However, it depends on the crop and island where the species were cultivated. In Santa Cruz, control is more efficient than Groasis Waterboxx in *Capsicum* and *Cucumis*, and the opposite with *Solanum*. The first two species responded better to the use of supplied water. However, the case of *Cucumis* productivity is interesting as on Floreana Island, Groasis Waterboxx^®^ is more efficient, showing that WSTs have a different response depending on the island as soil, quality of water and climate are different. Therefore, the WST effectiveness can be higher than control, depending on the crop and environment. In a similar pilot study, Jaramillo and colleagues (2015) detected an increased average growth rate in 22 species planted with Groasis Waterboxx^®^ WST over the control. 

The number of fruits varies based on season, crop and treatment. The number of fruits produced per plant is higher in the dry than in the wet season, as in productivity. We also observed that Groasis Waterboxx^®^ has a better effect in wet than in dry seasons based on the number of fruits. In the case of *Cucumis*, a crop present on both islands, we observed that the number of fruits was higher with control than the WST in Santa Cruz and Floreana. However, we highlight that in *Solanum* in the wet season, the number of fruits was much higher (50%) than the control in Santa Cruz. Thus, Groasis Waterboxx^®^ has an important effect on the production of fruits in this species on this island; however, in Floreana, control was more effective in the same species.

The IFW was higher in the wet than in the dry season. The IFW is higher as the fruits are affected by air humidity and more soil moisture, as confirmed in other studies [25,26]. In general terms, Groasis Waterboxx^®^ performed better than control in the dry season, showing that this WST produces more biomass per fruit unit. This is an advantage for farmers as bigger fruits have better profits in the local market. However, considering the island and crop, we found that these results vary considerably with lower differences in Santa Cruz and higher differences in Floreana, showing that the individual fruit biomass is affected by the different environmental conditions between the islands. 

The results of the effect of WST, such as Groasis Waterboxx^®^, may vary depending on environmental factors (e.g., temperature and precipitation). However, more studies are needed to confirm this finding.

As each year has different temperature and precipitation conditions, it was important to evaluate the differences in productivity between the years when the study was carried out. We found that in 2016 precipitation might affect productivity in the control treatment. Due to drought, Farmers had to use brackish water with high electric conductivity (6.8 dS/m) to irrigate their crops. However, in the crops where Groasis Waterboxx^®^ was used, productivity was much higher (57%) because this WST does not depend on irrigation. The device holds fresh water added during plantation and also collects rainwater. Therefore, drought and the high salinity from low-quality water for irrigation did not affect these crops. In 2017 (annual precipitation: 419.3 mm), the rainfall was higher than in 2016 (257.8 mm). We considered this year with average precipitation conditions (historical rainfall: 404.1 mm). Therefore, we did not see substantial differences between treatments (Figure 4). In 2018, the rainfall (362.4 mm) was lower than in 2017; however, productivity was higher in both treatments. Even though in 2018, the productivity was 22% higher in control than in Groasis Waterboxx^®^, we observed that productivity in 2018 within treatments and control was 77% higher than in 2016, and with Groasis Waterboxx^®^, the productivity in 2018 was 15% higher than 2016. Therefore, productivity was more balanced using Groasis Waterboxx^®^. This finding is crucial as using WST, such as Groasis Waterboxx^®^, may reduce the risk of low productivity during extreme climate conditions, giving farmers more security for harvesting [27].

The results linked with the effect of treatments on maximum plant height reached by the plant show that for *Capsicum* in Santa Cruz and *Cucumis* in Floreana, the use of Groasis Waterboxx^®^ does not differ statistically from control. In contrast, we found statistical differences in *Solanum* in both islands. Showing that the plant height is influenced by the WTS, depending on the island conditions and the species. In a similar study [4], the use of Groasis Waterboxx^®^ showed different performances depending on the species in Floreana and Santa Cruz Islands.

Our results confirm that plant height correlates well with productivity in two species: *Capsicum* and *Solanum*, showing that larger plants reflect higher productivity despite the applied treatment. Similar results show a positive correlation between productivity and plant height for other crops, such as cultivated lettuce [28] and maize [29,30]. However, it did not occur for *Cucumis*, where productivity is not significantly correlated with plant height in both treatments. Opposite results were found in a similar study [31] where plant height was highly correlated with crop production. 

This study enabled farmers to increase their understanding of WSTs and to recognize their potential, especially in breaking the reliance on brackish water for irrigation. However, despite farmers’ interest in the technologies, they expressed technical and economic obstacles and also structural and social limitations, evidenced by their constant references to the lack of support from the local government to solve urgent problems in the agricultural sector and to the low profitability of agriculture in the Galapagos, which translates into reduced profit margins and erratic work schedules [22]. In this regard, the unanswered question is whether farmers will adopt the new practice on a larger scale as an integral part of their farming system. From this study, it is difficult to determine whether their participation in the research will permanently change irrigation practices. However, it may be an essential first step to reconsider their current practices and move toward transition. Furthermore, the study shows that influencing sustainable changes in farming practices is a complex task that does not depend solely on good technical data on the benefits and efficiency of new technologies or increased farmer knowledge, but requires changes at the societal and institutional levels. Therefore, future research must look beyond the farm and involve other stakeholders in meeting farmers’ needs to unlock the potential of using water-saving technologies in Galapagos agriculture.

## 4. Materials and Methods

The research project revolved around two fundamental aspects: (a) participatory experiments to test the efficiency of the WST and (b) a transversal process of careful and continuous observation of the participant’s responses during the experimental phase. Working closely and communicatively with the participants during the research process was essential to register their responses. Therefore, initial participatory workshops were held to involve farmers in WST use and importance, defining specific aspects of the crops to be used where farmers’ decisions were respected and considered. Similarly, during the experimental phase, regular meetings were conducted to discuss the farmers’ experiences with WSTs, and address their concerns and the perceived barriers to using the technologies. Finally, after closing the data collection process, meetings were conducted to evaluate farmers’ perceptions of the whole study and gather any final comments, suggestions and opinions.

### 4.1. Selection of Participants

The selection of participants is essential in any participatory project [32]. For the present study, researchers contacted the potential participants at the free fair and invited them to participate in the project. In this space, local farmers offer their products to the community. After the initial contact, we visited these farmers on their farms and introduced the project idea and the potential of water-saving technologies (WST). During those visits, we carefully sought farmers interested in sustainability and the proposed technologies to be committed despite their busy working life. Working with farmers who voluntarily agreed to participate in the research process was essential for this study since interest and engagement are prerequisites for successful participative research [28]. Furthermore, this demonstrates an innovative capacity in farmers, crucial for exploring possible sustainable transitions [33]. No payment was made to cover the farmer’s working hours, but in some cases, specific support material for the experiments was provided, such as seedlings.

### 4.2. Study Sites 

Experiments were conducted in agricultural areas located in Santa Cruz and Floreana islands in different climatic zones from 2016 to 2018 (Figure 6). On both islands, annual precipitation increases with altitude [34,35,36]. Usually, monthly rainfall totals are higher during the wet-warm season than during the dry-cold season (Appendix A). The information on Galapagos soils is minimal, and there is almost no information on Floreana soils [37]. For Santa Cruz, it is known that the soil in the higher zones is deeper brown soil [13,38].

Floreana is the smallest inhabited island in the Galapagos archipelago and has a population of about 150 people [39,40]. Freshwater is very limited on the island. During the dry season, it is common to have water restrictions and consumption cuts. The population relies primarily on three aquifers with water flows that fluctuate seasonally according to rainfall [41]. Agricultural activity is carried out in the traditionally-called transition zone and the humid area. However, some farmers and families are settled in Floreana island’s arid zone and rely on small familiar orchards for food. Experiments in Floreana were carried out primarily near the town in the island’s dry area (15 to 24 m.a.s.l.). Two producers were in the humid agricultural zone (337 to 364 m.a.s.l.).

Santa Cruz Island is the trading and economic center of the Archipelago and the most populated island. Experiments in Santa Cruz were planted in the island’s transition zone (in an agroecological system known as evergreen seasonal forest and shrubland and humid highlands); the altitude varied from 160 to 440 m above sea level.

### 4.3. Experimental Process

The participatory experimental process consisted of four activities, (1) crop selection, (2) implementation of a water-saving technology in farms such as Groasis Waterboxx^®^, (3) watering and (4) monitoring of productivity considering the fruit weight and the number of fruits.

#### 4.3.1. Crops Selection

An essential part of the development of the experiment was farmers’ active involvement in deciding which crops to work on. The farmers themselves selected crops during a participatory workshop where they were encouraged to analyze, based on their experiences which crops they considered had the most significant commercial value and relevance to their livelihoods and to select accordingly to give value to participants time [41].

#### 4.3.2. Water-Saving Technology (WST)

One water-saving technology (WST) was used: Waterboxx^®^ (Groasis B.V., Franseweg, The Netherlands). The Groasis Waterboxx^®^ consists of a circular bucket of 15 L capacity that collects rainfall as well as water from dew and that transfers water from the bucket to the root system through a nylon wick by capillarity [42]. The Groasis Waterboxx^®^ has two evaporation covers, one at the bottom of the bucket in contact with the soil and a second at the top [5]. For controls, *Solanum*, *Capsicum* and *Cucumis* were planted 30 cm apart. For Groasis Waterboxx^®^, two plants per bucket were placed 15 cm apart.

Seedlings were randomly assigned to treatments, and farmers decided how many experimental units to use in the layout (Table 2). Field experiments were conducted from 2016 to 2018, contrasting Groasis Waterboxx^®^ versus control. Each specimen of *Capsicum*, *Cucumis* and *Solanum* was considered an experimental unit. Farmers tested WST based on their schedule as space and plant availability allowed. Finally, farmers decided in which crop or crops to test the technologies. 

#### 4.3.3. Watering Schedule

An initial amount of water of 20 L was supplied to both control and Groasis Waterboxx^®^ treatments at the time of planting. For the control treatments, all the water was poured over the soil, and for the Groasis Waterboxx^®^ treatments 5 L were poured over the soil and 15 L into the propylene bucket at the beginning of the experiment.

Plants assigned a control treatment (drip-irrigated) were watered according to the farmer’s schedule after being planted. The drip irrigation systems generally consisted of a water pump connected to reservoirs with pipes of variable diameter. The ball valve was connected by PVC pipes and coupled with 2 cm—diameter polyethylene drip lines parallel to the crop row [43,44]. At the start of the experiment, total irrigation using these systems was calculated by measuring the time to fill a 500 mL bucket, and final irrigation-water use was estimated at 0.3 L/plant /day. Farmer’s watering schedule usually consisted of two irrigation events per day, one in the morning and one in the afternoon, each with an approximate duration of 20 min. When it was not possible to use drip irrigation systems for any reason, farmers relied on manual irrigation. Manual irrigation consists of watering plants with water carried in buckets from reservoirs where rainfall water is mixed with brackish ground/crevice water. In drought years such as 2016 (annual rainfall 257.8 mm), farmers relied on purchased brackish water extracted and sold by privately-owned tankers for irrigation. This water had low quality, with levels of EC as high as 6.8 dS/m.

Plants assigned a Groasis Waterboxx^®^ treatment were not watered after being planted. They depended solely on the initial water and the water collected from rainwater and dew by the Groasis Waterboxx^®^. Therefore, when using this technology, farmers saved approximately 0.3 L/plant/day in irrigation water. For the crops used in this experiment, which usually complete their life cycle in 4–5 months (Appendix A), these savings could add up to 37.8–45.8 L per plant. Twenty-one farmers, fourteen from Floreana and seven from Santa Cruz, voluntarily enlisted to explore the potential of WST (Table 2).

#### 4.3.4. Monitoring of Productivity

A digital registration platform was developed to facilitate data registration as a first step in the monitoring process. Farmers were trained on the correct use of the tool and how to register the required data. As part of this training, farmers practiced reporting data with the supervision of the researchers to ensure the collection of good quality data.

Farmers monitored the crops during harvests. Monitoring varied based on crop harvest time. Farmer participants and the number of plants per farm and treatment are detailed in Table 2. Direct communication between farmers and researchers was maintained regularly to determine when fruits were ready for harvest. Harvests usually happened on Thursdays because, on Fridays, the farmers would gather their products to sell them to ships, and on Saturdays, they would sell them at the free fair.

The number of fruits was registered every time the crops were monitored. Usually, the crops were monitored every two weeks, but this varied according to species. The fruit weight was measured in kilograms at crop harvest. The number of fruits counted consisted of 7301 individual fruits corresponding to 2570, 286 and 4338 fruits of *Capsicum, Cucumis* and *Solanum*, respectively. The number of fruits measured per individual varied according to the crop species. For *Capsicum* and *Solanum,* the weight of 4–10 fruits per plant was recorded. For *Cucumis*, the weight of 2–3 fruits per plant was recorded.

#### 4.3.5. Data Analysis

To assess the efficiency of the water-saving technology: Groasis Waterboxx^®^, the following traits were selected in the studied crops (*Capsicum*, *Cucumis* and *Solanum*): Total plant productivity in kg, the total number of fruits produced and average individual fruit weight (IFW) in kg.

The assessment of WST was classified by islands: Santa Cruz and Floreana. In Santa Cruz, the study was carried out on three of the selected crops, whereas, in Floreana, only two (*Cucumis*, *Solanum*) crops were evaluated. 

Another important variable that we considered was the season as precipitation and temperature are different between two periods: wet-warm season from January to June, and cold-dry season from July to December.

The effect of Groasis Waterboxx^®^ as water-saving technology and the control (drip irrigation) on the studied traits was assessed using linear mixed models (LME), divided in seasons (Wet and Dry) In these models, the studied traits were treated as response variables and the treatments (Groasis Waterboxx^®^ plus control) as explanatory variables. As study sites were located in different zones (Humid, transition and arid), this variable was considered a random factor. 

As a post hoc test, Tukey HSD was applied between the treatments to reveal significant differences (multcomp package; [45]) with p values adjusted by the method of Holm–Bonferroni [46]. All statistical analyses were carried out using the R version 3.6.2 [47].

## 5. Conclusions

Using the WST Groasis Waterboxx^®^ can improve crop production and water use efficiency in Galapagos farms. Nevertheless, the benefits of Groasis Waterboxx^®^ depend on crop species, season and farm location.

Participatory research testing WST’s efficiency is an essential approach for increasing the understanding of the use and benefits of these technologies in the Galapagos. However, ensuring farmers follow the experimental process is challenging for this type of research.

Transitioning to a sustainable agriculture system in the Galapagos will require several changes in addition to recognizing the benefits of WST. The perceived cost of switching to such practices remains one of the main barriers.

## Figures and Tables

**Figure 1 plants-11-02848-f001:**
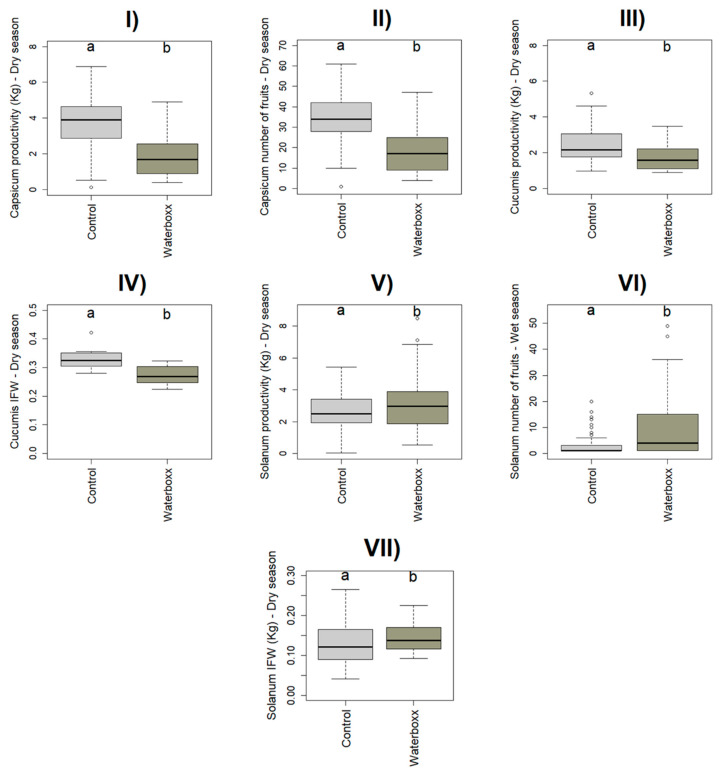
Significant effect of WST on crop’s plant traits from Santa Cruz represented by boxplot, showing the data distribution with maximum and minimum values, lower and upper quartiles, the median and outliers represented by dots outside the boxplot (same for Figure 2, Figure 3 and Figure 4). *Capsicum:* (**I**) Productivity-dry season, (**II**) Number of fruits—dry season; *Cucumis:* (**III**) Productivity—dry season, (**IV**) IFW—dry season *Solanum:* (**V**) Productivity—dry season, (**VI**) Number of fruits—wet season, (**VII**) IFW—dry season. Letters a and b represent significant letters.

**Figure 2 plants-11-02848-f002:**
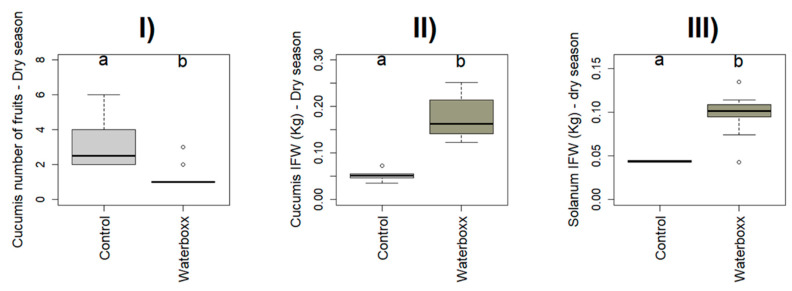
Significant effect of WST on crop’s plant traits from Floreana: *Cucumis:* (**I**) Number of Fruits—dry season, (**II**) IFW—dry season; *Solanum*: (**III**) IFW—dry season. Letters a and b represent significant letters. Letters a and b represent significant letters.

**Figure 3 plants-11-02848-f003:**
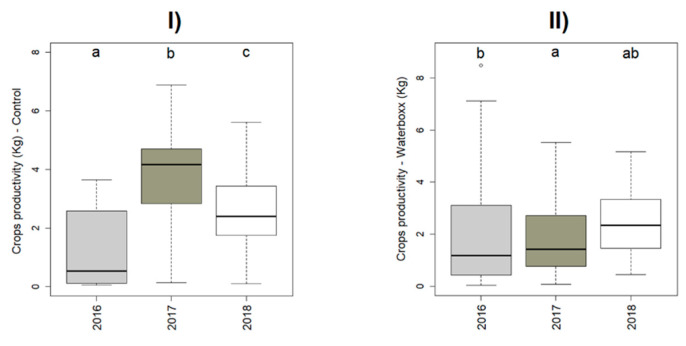
Effect of the research year on crop’s productivity, differentiated by treatment. The data are merged with *Capsicum*, *Cucumis* and *Solanum* information from both islands (Floreana and Santa Cruz). (**I**) Control treatment, (**II**) Groasis Waterboxx^®^ treatment. Letters show significant results among years within treatments.

**Figure 4 plants-11-02848-f004:**
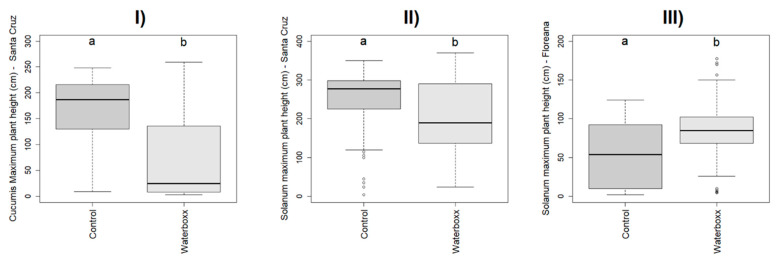
Significant effect of treatments on crop’s maximum plant height in cm. (**I**) *Cucumis*—Santa Cruz, (**II**) *Solanum*—Santa Cruz, (**III**) *Solanum* Floreana. Letters a and b represent significant letters.

**Figure 5 plants-11-02848-f005:**
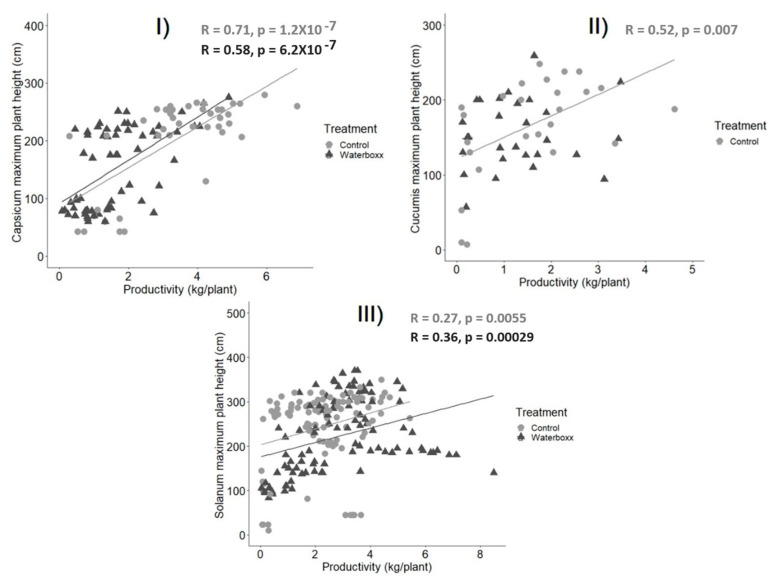
Pearson correlation between maximum plant height reached by the plant in cm and total productivity in kg comparing both treatments: Groasis Waterboxx^®^ and control. (**I**) *Capsicum*, (**II**) *Cucumis*, (**III**) *Solanum.* The data of *Cucumis* and *Solanum* are merged between both Islands (Santa Cruz and Floreana). Only significative values are shown.

**Figure 6 plants-11-02848-f006:**
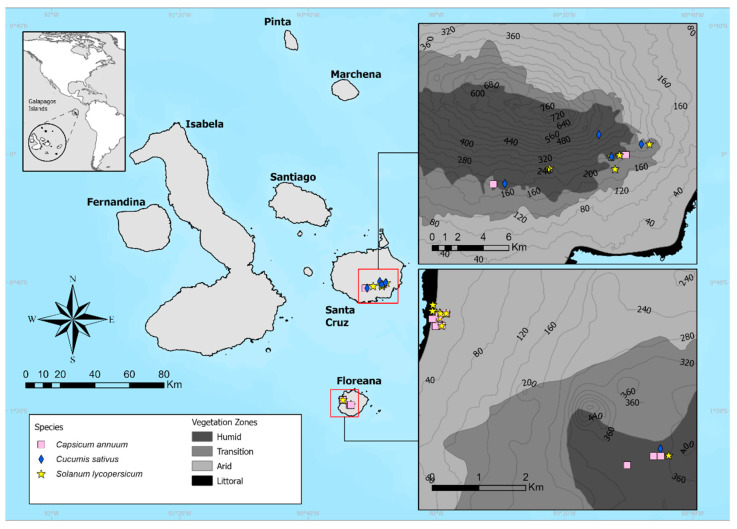
GPS coordinates where the farms from Santa Cruz and Floreana island are located, the map displays the agriculture zones within islands also the type of crop that each farm produced.

**Table 1 plants-11-02848-t001:** Variability within plant traits based on the effect of treatment (WST, control). Values in bold are significant. Values in blank mean no productivity or deficient data on that season.

Santa Cruz Island												
	Plant Productivity				Fruit Number				IFW			
Season	Wet		Dry		Wet		Dry		Wet		Dry	
Crop	χ^2^	*p* value	χ^2^	*p* value	χ^2^	*p* value	χ^2^	*p* value	χ^2^	*p* value	χ^2^	*p* value
*Capsicum annuum*	0.0952	0.7577	**42.460**	**<0.001**	5 × 10^−4^	0.981	**41.863**	**<0.001**	5 × 10^−5^	0.999	0.0661	0.797
*Cucumis sativus*	0.196	0.657	**4.006**	**0.045**	0.017	0.895	1.109	0.292	1.136	0.286	**22.567**	**<0.001**
*Solanum lycopersicum*	0.479	0.488	**6.336**	**0.012**	**19.183**	**<0.001**	0.0087	0.9257	0.616	0.432	**8.2463**	**0.004**
Floreana Island												
	Plant Productivity				Fruit Number				IFW			
Season	Wet		Dry		Wet		Dry		Wet		Dry	
Crop	χ^2^	*p* value	χ^2^	*p* value	χ^2^	*p* value	χ^2^	*p* value	χ^2^	*p* value	χ^2^	*p* value
*Cucumis sativus*	-	-	1.664	0.197	-	-	**9.344**	**0.022**	-	-	**41.143**	**<0.001**
*Solanum lycopersicum*	-	-	0.240	0.623	-	-	1.401	0.236	-	-	**9.8319**	**0.001**

**Table 2 plants-11-02848-t002:** Location and number of plants of each of the farmers and family garden who participated in the on-farm evaluation (2016–2018), using water saving technologies in five cultivated species.

Farmers	Farms and Familiar orchards	Island	Latitude	Longitude	Altitude m.a.s.l.	Total Plants	Species	Groasis Waterboxx	Control
F1	Agriculture Association	Santa Cruz	−0.661284	−90.293393	407	20	*Cucumis sativus*	10	10
F2	Farm Wilson Cabrera	Santa Cruz	−0.675676	−90.278887	250	172	*Capsicum annuum*	43	43
							*Solanum lycopersicum*	43	43
F3	Farm Nixon López (Cascajo)	Santa Cruz	−0.685565	−90.281964	225	76	*Solanum lycopersicum*	38	38
F4	Farm Nixon López (Occidente)	Santa Cruz	−0.695952	−90.359384	200	120	*Capsicum annuum*	30	30
							*Cucumis sativus*	30	30
F5	Farm Santa María	Santa Cruz	−0.676825	−90.284441	400	26	*Solanum lycopersicum*	6	6
							*Cucumis sativus*	7	7
F6	Farm Teodoro Gaona	Santa Cruz	−0.685192	−90.327934	252	16	*Solanum lycopersicum*	8	8
F7	Familiar orchards Glenda Peña	Santa Cruz	−0.668071	−90.263588	450	16	*Solanum lycopersicum*	4	4
							*Cucumis sativus*	4	4
F8	Familiar orchards Cecilia Salgado	Floreana	−1.274631	−90.486448	20	2	*Solanum lycopersicum*	1	1
F9	Farm Claudio Cruz	Floreana	−1.30399	−90.449874	350	10	*Capsicum annuum*	5	5
F10	Farm Francisco Moreno	Floreana	−1.301857	−90.443452	363	78	*Capsicum annuum*	13	13
							*Solanum lycopersicum*	13	13
							*Cucumis sativus*	13	13
F11	Familiar orchards Aníbal Altamirano	Floreana	−1.273122	−90.487181	21	2	*Solanum lycopersicum*	1	1
F12	Familiar orchards Iván Altamirano	Floreana	−1.302265	−90.443386	19	2	*Capsicum annuum*	2	2
F13	Familiar orchards Jose Mora	Floreana	−1.275648	−90.485828	28	4	*Capsicum annuum*	1	1
							*Solanum lycopersicum*	1	1
F14	Familiar orchards José Naula	Floreana	−1.275052	−9.048609	16	4	*Capsicum annuum*	2	2
F15	Familiar orchards Aura Cruz	Floreana	−1.274319	−90.487262	15	2	*Solanum lycopersicum*	1	1
F16	Familiar orchards Carmen Mora	Floreana	−1.274631	−90.485865	27	6	*Solanum lycopersicum*	3	3
F17	Familiar orchards Flor Naula	Floreana	−1.27481	−90.485674	20	2	*Solanum lycopersicum*	1	1
F18	Familiar orchards Lelia Cruz	Floreana	−1.275849	−90.487346	18	2	*Capsicum annuum*	1	1
F19	Familiar orchards Luz María Mora	Floreana	−1.277249	−90.486655	18	2	*Capsicum annuum*	1	1
F20	Familiar orchards Mayra Gómez	Floreana	−1.274764	−90.484622	15	4	*Capsicum annuum*	1	1
							*Solanum lycopersicum*	1	1
F21	Familiar orchards Vilma Pérez	Floreana	−1.277152	−90.485468	24	20	*Capsicum annuum*	5	5
							*Solanum lycopersicum*	5	5
								**294**	**294**

## Data Availability

Further information is available at: https://gv2050.shinyapps.io/GV2050-restoR/; http://www.galapagosverde2050.com/ (accessed on 30 August 2022).

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
