# Peer review of "Adoption of Sustainable Agriculture Practices through Participatory Research: A Case Study on Galapagos Islands Farmers Using Water-Saving Technologies"

_plants, 2022, doi:10.3390/plants11212848_

Round 1

Reviewer 1 Report

The aim of this study was to test the effectiveness of 14 using WSTs in five of the most valuable crops in the islands, through participatory research with 15 the involvement of a group of farmers from the Floreana and Santa Cruz islands, and explore a 16 possible transition to more sustainable agricultural practices. The aim were clear, however, there are several flaws (list following) of the study that make me have to reject it for publishing in this journal.

1)       When are these selected species’ growing stages? And which season is the WST applied? All year long, or only one season? I can see from the Figure S4 and S5, there are wet and dry seasons on both islands, apply the WSTs during different seasons may vary a lot, thus, there may be risky for the authors to put all the data together to analyze the effects of WSTs.

2)       The authors wanted to evaluate the effects of the WSTs, however, the water saving effects were not monitored, which should be an important aspects for these technologies. And with these data missing, it is impossible to do the profit analysis (input and output balance) to give a full evaluate of those technologies.

3)       In the section 4.3.2, the authors mentioned that the study were conducted from 2013 to 2018. But I am not sure if it means all the farmers participate all the years, or some farmers participate certain years, others participate other years? In other words, the sample numbers in Table S1 are the same for all the years or are these the total numbers for all the years?

4)       In the section 4.3.4, the authors mentioned that the crops were monitored depends on the phonological stages, then which stage were the plant height data showed in Table S2 and the statistics showed in Table 1? Or, the average of all the stages?

5)       In drought years such as 2016 (annual rainfall 579.8 mm), farmers relied on purchased brackish water for irrigation. This water had low quality with levels of EC as high as 6.8 dS/m. Such high EC brackish water may influence the water saving effects of the WSTs, these data should be separated analyzed.

Other minor suggestions,

1)       Some tables and figures are important enough to be put into the text instead of supplementary materials, such as Table S1

2)       There are two figure 1 in the manuscript.

3)       I didn’t see the necessary of the section 2.1, maybe it should be put in the introduction?

4)       Line 132-139, there is no need to keep three digitals for “cm”

5)       Line 147, the word “different” should be more specific as higher or lower.

6)       I am not sure why the authors show the SRT or CRT numbers in figure 1 instead of the original values. The same problems with QT in figure 2. And I believe that there is something wrong with the explanations for those abbreviations in section 4.3.5, such as the SRT in line 416.

7)       Line 227-228, the productivity were 16% more and the fruits were 42% more, does that means the fruits are smaller than control? And I am not sure is the smaller fruits sell at much less price, if it is true, this should be considered in the profit analysis.

8)       Line 230-231, this sentence does not make sense.

9)       Section 4.3.3. why there is “At the start of the experiment,…” then what?

Reviewer 2 Report

Dear Editor, in the manuscript Plants-1920729 authors aimed to evaluate the effectiveness of farmers water-saving technologies (WSTs) in Galapagos Archipelagoby using five of the most valuable crops in the islands (Brassica oleracea, Capsicum annuum, Citrullus lanatus, Cucumis sativus, and Solanum lycopersicum), through participatory research with the involvement of a group of farmers.

In general the manuscript provides interesting information and could be suitable for publication after minor revision:

- Avoid using expressions of personal thinking, such as “authors believe that…..” Statements should be based on the obtained results.

- Section 3: It is not necessary to used subheadings in this section.

- Try to summarize the main conclusions of the experiment in a separate section or in a last paragraph, according to the journal format.

Round 2

Reviewer 1 Report

COMMENTS to the manuscript:

TITLE: Adoption of Sustainable Agriculture Practices Through Participatory Research: A Case Study on Galapagos Islands Farmers Using Water-Saving Technologies

The authors responded most of my comments, now the manuscript looks better, but still need minor revisions.

1)      There are still two figure 1. Please check that carefully!

2)      It is better to rearrange figure 1, 2 and 4, put three small figures in a line. And you need to give the explanations of what are the dots, boxes, bars and lines in the figure represent.

3)      The subtitle of 2.2.4, is it better to change “the year” into research years?

4)      Figure 3 the X axis should be “Treatments”. And the explanation of letters in the figure must be something wrong. “Letters of significant differences are among years within treatments?

5)      Figure 5, the X axis the units of the production should be kg/plant?  The legends should be given for the lines in the figures. The second figure (II), there is one line with the p values over 0.05, that line should be omitted.

6)      I am afraid the table 2 was not referred in the manuscript correctly.
